# Analyzing the Bonding Resistance of the Ribbed Stainless-Steel Bar in the Refractory Castable After High-Temperature Treatment

**DOI:** 10.3390/ma18061282

**Published:** 2025-03-14

**Authors:** Linas Plioplys, Andrius Kudžma, Valentin Antonovič, Viktor Gribniak

**Affiliations:** 1Laboratory of Innovative Building Structures, Vilnius Gediminas Technical University, Sauletekio Av. 11, LT-10223 Vilnius, Lithuania; linas.plioplys@vilniustech.lt; 2Laboratory of Composite Materials, Vilnius Gediminas Technical University, Linkmenu Str. 28, LT-08217 Vilnius, Lithuania; andrius.kudzma@vilniustech.lt (A.K.); valentin.antonovic@vilniustech.lt (V.A.)

**Keywords:** refractory castables, pull-out tests, heat treatment, finite element model, deformation modulus

## Abstract

Calcium aluminate cement-based castables were developed in the early 1990s for the metallurgical and petrochemical industries, exhibiting exceptional mechanical resistance when heated over 1000 °C. In typical operation conditions, they withstand compressive stresses due to high temperatures and mechanical loads. The extraordinary material performance has led to interest in using these materials for developing building protection systems against fires and explosions. This application requires structural reinforcement to resist tensile stresses in the concrete caused by accidental loads, making the bonding of reinforcement crucial. The different temperature expansion properties of the castables and reinforcement steel further complicate the bonding mechanisms. This manuscript belongs to a research project on developing refractory composites for civil infrastructure protection. In previous studies, extensive pull-out tests evaluated various combinations of refractories and reinforcement types to determine the most efficient candidates for refractory composite development. Thus, this study employs ribbed stainless Type 304 steel bars and a conventional castable, modified with 2.5 wt% microsilica for a 100 MPa cold compressive strength. It uses the previous pull-out test results to create a numerical model to predict the bonding resistance of the selected material combination. Following the composite development concept, this experimentally verified model defines a reference for further developing refractory composites: the test outcome of a new material must outperform the numerical prediction to be efficient. This study also delivers an empirical relationship between the castable deformation modulus and treatment temperature to model the reinforcement pull-out deformation in the composite heated up to 1000 °C.

## 1. Introduction

Designing refractory materials ensures their resistance to extreme temperatures, typically exceeding 1000 °C, and is essential in glass and cement production, metallurgy, and petrochemical industries [1,2,3]. Refractory concretes originated in 1955 in Japan; however, modern calcium aluminate cement-based refractory castables emerged in the 1990s [4], offering superior properties to traditional refractory materials. Over time, advancements in ceramics and synthetic binders have expanded their applications, making them indispensable in high-temperature operating conditions. Today, refractories line furnaces, reactors, and kilns, ensuring structural integrity and thermal resistance [5]. Despite the interest in industrial applications, traditional refractories are rarely reinforced with metallic meshes or steel bars because of challenges in achieving uniform stress distribution in the structures operating at high temperatures [6]: temperature-induced stresses can weaken the bonding resistance of the bar reinforcement in the refractory matrix.

Computer-aided modeling may provide a valuable tool for optimizing refractory formulations and predicting their performance under extreme conditions [7,8]. However, simulating interactions between aggregates, cement, and additives, accounting for thermal expansion, shrinkage, and phase transformations, and validating computational results are challenging because of the lack of reliable material models, test data, and the interaction of the temperature and mechanical problems [9,10]. The emergence of steel reinforcement further complicates the modeling issue [11].

Notwithstanding the above challenges, modern castables have drawn interest in fire- and explosion-resistant building protection systems [12,13,14]. However, such applications require structural reinforcement to withstand tensile stresses caused by accidental loads, making reinforcement bonding a critical factor [11]. One of the main challenges in reinforcing refractory concretes is the difference in thermal expansion properties between castables and steel reinforcement, which complicates their interaction and affects the overall mechanical performance of the structural composite [15,16,17,18]. Studies addressing this issue [12,13,19] have conducted extensive pull-out tests on various refractory and reinforcing material combinations. The results indicate that reinforcement bonding resistance improves significantly in low-cement castables. From a cost perspective, the financial efficiency of conventional and low-cement refractory materials in terms of pull-out deformation energy remains comparable, making conventional refractory materials viable alternatives to expensive low-cement castables.

In addition, the conventional castable (CC) materials, mixed on-site using refractory aggregates and calcium aluminate cement (CAC), offer several technological advantages, including easy blending, shaping, and mixture stability [20,21]. They are attractive due to their good pouring flowability, mechanical strength, and cost-effectiveness in moderate- to high-temperature environments. However, CC loses mechanical resistance when exposed to temperatures above 800 °C, with cold compressive strength (CCS) potentially decreasing by more than 50% [22] due to binder dehydration and cement mineral recrystallization [23]. Despite this limitation, CC remains widely used in industrial applications where strict mechanical performance requirements are not a priority.

The addition of microsilica (SiO_2_) can significantly enhance the mechanical performance of CC, tripling strength [24,25], improving thermal and abrasion resistance, and increasing sinterability temperature [26]. The previous testing program [12,13,19] also demonstrated that the strength of CC is insufficient to ensure the bonding resistance of steel bars after temperature treatment, and the plain bars lose their bonding contact with concrete independently of the concrete strength. These restrictions discredited the application of these materials to develop the refractory composite system. However, the stainless Type 304 steel ribbed bars and the conventional castable with 25 wt% CAC, modified with 2.5 wt% microsilica for a 100 MPa CCS (designated as CC_m_ in this study), were found to be promising for developing refractory composites.

The industry acknowledged the necessity for a systematic approach to innovation, design, and technology at the early composites’ development stage [27,28]. However, research at the organizational and operational levels of composite manufacturing is limited [29]. Despite significant advancements in composite material science, no systematic efforts have been made to address composite functionality and fabricability concerns. Particular composite development procedures are necessary when materials lack standardized functionality etalons. Although composite materials can be customized for specific applications, this flexibility brings challenges [30]: quality assurance and control are essential for achieving target performance characteristics and overall integrity. Chatzimichali and Potter [31] offered insights into the socio-technical factors influencing the diffusion of manufacturing innovations in composites. Still, academic research mainly focuses on composites’ chemical and physical properties, emphasizing design aspects like strength prediction and damage characterization [32,33]. Most studies target simple structural designs for specific properties such as bending resistance, buckling, impact minimization, and reduced mass [34,35]. Less focus has been placed on formalizing design for more complex geometries in commercially manufactured parts.

Computer-aided design with experimentally verified numerical models can provide a powerful tool for developing composites [33,36]. Garnevičius and Gribniak [37] created an adaptive design concept for developing functionally efficient structural systems, where the experimentally verified finite element (FE) model serves as a reliable reference for assessing system efficiency. According to this concept, the performance of a new material must exceed the numerical prediction to be considered efficient. This iterative design procedure ensures that the experimental outcome of a new (enhanced) material surpasses the numerical prediction by identifying the sources of disagreement between numerical simulations and test outcomes, minimizing the need for experimental trials and streamlining the composite development process.

Following the composite development concept [37], this study develops a reference FE model to develop refractory composites that protect civil engineering infrastructure from fire and explosions. It focuses on the reinforcement bonding resistance in refractory castables. This bond model forms a benchmark for further enhancing the mechanical performance of reinforcement systems (composites) under combined temperature and mechanical loads. This study builds on previous test results [12,13,19] from the same research project and integrates ribbed stainless Type 304 steel bars with the modified castable CC_m_ to develop the FE model since this combination of refractory materials ensures reliable bonding contact, which was chosen as the reference for developing refractory composites.

## 2. Experimental Database

This study employs the pull-out test data reported by Plioplys et al. [12,13,19]. The comprehensive experimental program included 247 castable samples for compression tests (70 mm cubes) and 252 pull-out specimens. In addition, six X-ray diffraction (XRD), four scanning electron microscopy (SEM), and three X-ray fluorescence (XRF) specimens are analyzed. This manuscript reports the testing characteristics and experimental results essential for understanding the modeling object. An interested reader can find detailed information in the open-access publications [12,13,19].

This study focuses on the stainless Type 304 steel ribbed bars and the CC_m_ with the target CCS of 100 MPa since this combination of refractory materials ensured a reliable bond with reinforcement, and previous studies [12,13,19] chose it as the reference for further developing refractory composites. This castable includes 70 wt% of fireclay aggregates (≤5 mm), 25 wt% of CAC Istra 40 (Calucem, Mannheim, Germany) with 40 wt% of Al_2_O_3_, 2.5 wt% of microsilica (SiO_2_), 2.5 wt% of the milled quartz sand (QS, JSC Anykščių Kvarcas, Anykščiai, Lithuania), and the polycarboxylate ester-based deflocculant Castament FS30 (BASF Construction Solutions GmbH, Trostberg, Germany).

This manuscript develops an FE model of the pull-out test of the 8 mm stainless-steel ribbed bars in CC_m_ 100 mm cubic samples. Figure 1a shows the reinforcement bars’ surface geometry. The pull-out setup, used in this study and shown in Figure 1b, employs a slightly modified testing layout proposed by Chu and Kwan [38]. The ISO 1927-5 standard [39] determined castable specimens’ curing, drying, and heating specifications. Thus, all the castable samples, i.e., 70 mm cubes for compression tests and specimens for pull-out tests, were demolded after 72 h of curing in laboratory conditions at 20 ± 1 °C and dried for another 72 h at 110 ± 5 °C using a 2.0 kW heating camera Snol 3.5 (Umega, Ukmergė, Lithuania). The reference specimens were tested without additional heat treatment. In contrast, alternative castable samples were additionally heated for five hours at 400 °C, 600 °C, 800 °C, and 1000 °C using a 3.4 kW furnace Snol 30/1100 with an electronic controller (Umega, Lithuania) and a heating rate of the 2.5 °C/min rate to 700 °C and 5.0 °C/min up to 1000 °C. The naked reinforcement samples were heated at the same temperature as the corresponding castable specimens. All the test samples were cooled in laboratory conditions to 20 ± 1 °C before the physical tests.

The tests were conducted using an H75KS electromechanical testing machine (Tinius Olsen, Redhill, UK) with a maximum load capacity of 75 kN and a position measurement accuracy of ±0.01%. It applied tensile deformation to the bar at a controlled 2 mm/min rate. A 50 kN load cell, boasting a measurement precision of 0.5%, captured the reaction force. Two linear variable displacement transducers (LVDTs) were employed to monitor the bar’s relative displacement, each offering an accuracy of 0.02%. The mean value of their readings was utilized for subsequent analysis. Figure 1b,c shows the pull-out specimen schematic and test setup. Data acquisition and processing were managed by an Almemo 2890-9 signal processing unit (Ahlborn Mess- und Regelungstechnik GmbH, Holzkirchen, Germany), which, in conjunction with a laptop computer, recorded and processed measurements from the LVDTs and the load cell at one-second intervals.

Plioplys et al. [12,13] described the materials’ characterization conditions. Figure 2 shows the compressive and ultrasonic pulse velocity (UPV) test results of 70 mm CC_m_ cubes and 400 mm long tensile samples of the 8 mm ribbed stainless-steel bars. The testing process involved reference (unheated) samples and four heating conditions (i.e., 400 °C, 600 °C, 800 °C, and 1000 °C). After the heat treatment, all the specimens were cooled in the laboratory and tested at 20 °C before the physical tests. Five tensile reinforcement samples and five CC_m_ compression cubes were tested for each temperature condition. The CCS tests were conducted after the UPV measurements, using the same cube specimens.

Four pull-out specimens were prepared and tested for all the heating temperatures. Figure 3 shows the pull-out test results expressed in the applied load (**P**) and measured bar displacement (*u*) terms, and Figure 1b shows the measurement schematic. Thus, Figure 1, Figure 2 and Figure 3 determine the modeling object and the characteristics of the constitutive models in this study.

## 3. Finite Element Model of Pull-Out Test

Numerous bar and concrete interaction models have been developed over the years, e.g., [40,41,42,43,44], each employing different approaches to nonlinear analysis. Due to this diversity, establishing universal guidelines for their application in engineering practice is impractical. Instead, recommendations must be tailored to specific constitutive models, individual modeling approaches, or software platforms. This paper does not aim to provide an exhaustive review of the existing models. Readers are referred to the investigation by Jirásek and Bažant [45] for a more comprehensive discussion. This study extends the previous pull-out modeling case [46] to simulate the bonding behavior of ribbed bars (Figure 1a) in refractory castable after high-temperature treatment. The example presented here was computed using the finite element (FE) software Atena (Version 2024n, Cervenka Consulting s.r.o., Prague, Czech Republic) [47]. As a result, some conclusions are specific to this software or, more broadly, to models based on the smeared crack approach and crack band method. These limitations primarily affect the bonding characteristics of the bars and castable.

These simulations employ a fracture–plastic material model [48,49]. Following this approach, the tensile concrete model employs the Rankine criterion with the exponential softening law by Hordijk [50]. The crack band model [51] relates crack opening displacements to fracture strains. Červenka et al. [52] determined a similar fracture approach for compression concrete. Menetrey and Willam [53] described the strength criterion of concrete plasticity under multiaxial stress conditions with nonlinear hardening/softening law. The modeling approach by Cervenka and Margoldova [54] adjusted the crack band and the crush band size, ensuring the dynamic crack orientation with the load.

This numerical study employs the perfect bond model because of its simplicity. It prohibits a slip over the contact surface of the steel reinforcement and castable material. Alternative modeling approaches to reinforced concrete may apply slip over the inter-material contact through contact elements, empirical models, or other simulation means [55,56,57]. These elaborate bond models require specific parameters, where experimental determination is not straightforward because of the local slip nonlinearities [58,59]. Moreover, the so-called bond–slip model complicates the iterative solution process, raising the simulation costs and limiting its practical value [41,60,61].

Previous numerical studies [62,63], which employed the Atena software, proved the adequacy of the perfect bond model for modeling reinforced concrete cracking and failure. Thus, this study also uses the perfect bond model to simplify the simulation concept.

### 3.1. Modeling Concept

Employing previous FE simulation concepts [41,46], this study considers the detailed model of the reinforcement ribs. However, such a simulation approach is computationally consuming [41]. Therefore, only a twelfth of the pull-out test sample (Figure 1b) was simulated. Figure 4a illustrates the fragmentation layout—the dark gray fragment indicates the modeled part of the sample. This model (Figure 4b) consists of approximately 55k finite elements, agreeing with the FE size recommended in the literature [41,64] for the 3D modeling problem of the ribbed bond contact. The ribbed surface (Figure 4c) equivalently represents the rib geometry (Figure 1a) measured with an electronic caliper at a precision of 0.01 mm. The ribbed 40 mm surface in Figure 4c corresponds to the bonding length in Figure 1b. The FE model assumes the perfect bond model for this zone, while no bond was accepted for the remaining part of the bar (Figure 4c).

The results of Figure 3 define the simulation object. The model’s boundary conditions (Figure 4b) correspond to the pull-out test (Figure 2b): the vertical and horizontal displacements of the top surface of the concrete were fixed, and the displacement increment was attributed to the top surface of the reinforcement bar; the 15 mm displacement was achieved in 75 increments.

Figure 2a indicates that the heating conditions did not significantly affect the CCS of CC_m_: overlapping the vertical error bars (representing the standard deviation) indicates the statistically insignificant difference in CCS. At the same time, the results of Figure 3 demonstrate the apparent dependency on the heating conditions. These results confirm that the compressive strength of the considered castable is not the sole control factor of the ribbed bars’ bonding resistance. This conclusion aligns with the findings by Senthil et al. [65], who found that increasing concrete strength over a certain level does not improve bonding resistance. Thus, the simplified material models neglect the decrease in the steel strength (Figure 2b) and assume that the deformational modulus of concrete (*E_c_*) controls the bonding resistance, reflecting the high-temperature treatment effects.

### 3.2. Modeling Pull-Out Resistance

The results of Figure 3 are scattered and hardly assessable. Therefore, Plioplys et al. [13] suggested the pull-out energy approach to quantify and compare the pull-out assessment results. Following this concept, the area under the load–deformation diagram (Figure 3) determines the deformation energy (*E_u_*); the 15 mm displacement terminates the energy-assessing area to ensure estimation equivalency. Figure 5a shows the schematic that describes this energy estimation concept.

The first modeling approximation assumes the deformation modulus values calculated using the UPV (Figure 2a) and material density (*ρ*) measurement results (Table 1) and the following empirical equation [66]:(1)$$E_{c}={UPV}^{2}\cdot \rho \cdot \left(1+\nu \right)\left(1-2\nu \right)/\left(1-\nu \right),$$
where *ν* is Poisson’s ratio (equals 0.17 and independent of the temperature treatment). Figure 5b shows the calculation results (blue bars) and Table 1 specifies the remaining material parameters of the FE model.

As expected, the pull-out energy assessment results defined substantial errors since the estimated UPV (Figure 2a) does not demonstrate the reduction tendency characteristic of the pull-out diagrams (Figure 3) after the temperature treatment. The deformation modulus (*E_c_*) correction was performed in two stages. In the first stage, the deformation modulus *E_c_* was tailored to reflect the pull-out energy released in the reference specimens and the samples heated at 800 °C and 1000 °C. In the second stage, the nonlinear regression model approximated the deformation modulus values corresponding to 400 °C and 600 °C heating temperatures, and the resultant pull-out energies are assessed. Figure 6a shows the corresponding regression model, which relates the heating temperature and deformation modulus of CC_m_. This diagram’s reference conditions correspond to the castable samples’ initial drying at 110 °C [12,13]. Figure 6b shows the pull-out energy assessment results. This figure demonstrates the remarkable agreement between the experimental pull-out energy estimation results and the numerical prediction outcomes.

For comparison purposes, the modified deformation modulus values (Figure 6a) are also distributed with the UPV calculated results in Figure 5b. Although the mean error of the UPV and Atena estimated modulus ratio does not exceed 5%, the UPV measure did not capture the temperature effect apparent for the pull-out test results (Figure 6b). The bar reinforcement, particularly the different temperature expansion properties of steel and castable, explains the observed decrement in the bonding resistance [8,69]. The following simple example reveals the unfavorable expansion effect of steel reinforcement.

### 3.3. Analyzing Uneven Expansion of Materials

This simple test demonstrates the effect of different expansion characteristics of the steel bar and refractory castable. For this purpose, a 70 mm cube from CC_m_ with a 40 mm long fragment of the ribbed stainless-steel bar was treated at 1000 °C. The sample fabrication and heating procedures were the same as those used in the pull-out tests (Section 2). Figure 7a shows the corresponding FE model. This model does not resist external mechanical loads. Therefore, the ribs (e.g., Figure 4c) were not modeled for simplicity. All remaining parameters were the same as assumed in Section 3.2 and determined in Table 1, including the perfect bond model over the castable contact with reinforcement. The FE model assumes the 24.3 GPa modulus of elasticity, corresponding to the considered heating conditions (Figure 5b). The prescribed deformations applied in 75 increments simulate the temperature expansion effect. These deformations correspond to the heating temperature (1000 °C) and the temperature expansion coefficients in Table 1.

Figure 7b shows the splitting cracking results observed during the physical test. Although the experimental sample was observed after cooling, this figure reveals the evident cracking around the bar. The predicted cracking shape (Figure 7c) reflects the experimental crack distribution, with a maximum predicted crack of 0.37 mm in the test specimen heated at 1000 °C. This cracking indirectly indicates the substantial effect of the temperature expansion on the loss of bond integrity.

## 4. Discussion of Results

The comprehensive pull-out tests [12,13,19], which included over 200 samples of different refractory castables and reinforcing materials, determined the material choice for developing reinforced refractory composites. The mechanical performance and material costs defined the selection of the castable; still, the ribbed stainless Type 304 steel bars were a single candidate because of the efficient combination of outstanding mechanical performance under high-temperature conditions (Figure 2b) and bonding characteristics resulting from the ribbed surface. Figure 8 shows the pull-out modeling results, determining the analysis object. In this context, the numerical modeling conducted in this study revealed the following essential aspects.

### 4.1. The Modeling Adequacy Analysis

This numerical study and previous physical tests [12,13,19] belong to the research project developing the refractory composite, which combines CC_m_ and ribbed stainless Type 304 steel bars. Typically, refractory castables have no structural reinforcement and no bond models are necessary. This study is dedicated to filling this modeling gap.

Figure 5b shows that the UPV counts may reasonably estimate the deformation modulus and approximate the modeling parameters. The temperature expansion deteriorates the bonding resistance, and Figure 7b illustrates the deterioration (splitting cracks) caused by the temperature-induced expansion of the steel bar. However, notwithstanding the simplicity of the modeling assumptions, i.e., the deformation modulus of the concrete simulates the adverse temperature effects on the bond strength, the proposed model adequately predicted the mechanical resistances observed during the pull-out tests. Except for the highest temperature (Figure 8e), where the non-elastic deformations of steel (Figure 2b) become non-negligible, the modeling results of Figure 8 adequately capture the shape of the test diagrams.

The pull-out energy predictions (Figure 6b) further prove the adequacy of the modeling concept (Figure 4c) in predicting global (deformation) bond behavior: the average prediction error does not reach 4%. This conclusion aligns with the previous findings on the adequacy of the perfect (no-slip) bond concept [62,63]. The standard deviation of the experimental pull-out deformation energy assessments (Figure 6b) varies from 11.7% to 22.4% depending on the heating temperature with the 16.2% average value. These results are typical for cementitious composites [38,70]. Thus, the obtained model can be considered experimentally verified and reliable for estimating the deformation modulus of the selected refractory castable reinforced with the stainless-steel ribbed bars.

The CCS (Figure 2a) controls the mechanical resistance of the bond (Figure 3), but this parameter alone is insufficient to predict bond resistance through the FE modeling approach. For the selected simplified geometry (Figure 4c) with the perfect bond model, the “effective” concrete strength of the CC_m_ (controlling the bonding resistance) could reach 25.5 MPa. This value represents a quarter of the CCS (Figure 2a). However, it nominally corresponds to the maximal bond stress [71].(2)$$\tau_{max}=2.5\cdot \sqrt{f_{c}},$$
where the compressive strength (*f_c_*) is equivalent to CCS. This equation delivers a physical rationale for the simplified bond model.

The ASTM methodology [66] is adequate for estimating the deformation modulus (*E_c_*) of the selected CC_m_ (Figure 5b). Figure 9 shows the corresponding difference between the deformation modulus of CC_m_, estimated using the UPV measurements, and the pull-out energy estimation results. The black whiskers show the experimental results’ scatter (standard deviation).

The absolute average error in Figure 9 does not exceed 6 GPa or 12% of the reference modulus value. After the temperature treatment, these differences demonstrate a distinct trend when the difference Δ*E_c_* changes from 12.6% UPV-based underestimation at 400 °C to 18.8% overestimation at 1000 °C. In the intermediate temperature range, the deformation modulus prediction differences are insignificant. Thus, the UPV-based estimations can be used for FE modeling of the selected castable. However, the bonding resistance of steel bars in refractory composite decreases after high-temperature treatment (Figure 3). Therefore, this study proposes the empirical equation for the deformation modulus of concrete to represent the adverse temperature effect on the bonding resistance as follows(3)$$E_{c}=86.56-8.89\cdot \ln t,$$
where *t* is the treatment temperature in °C and *E_c_* is in GPa. The alternative modeling approach could include the temperature expansion factor (e.g., Section 3.3). However, this analysis may require an elaborate interaction of temperature and mechanical analyses.

### 4.2. The Bond Deterioration Mechanisms

Refractory castables undergo microstructural transformations due to dehydration and phase transformations under high temperatures. As water is expelled from the hydrated phases, the material experiences shrinkage, increased porosity, and cracking [24]. These changes compromise the castable’s overall mechanical integrity and resistance. The primary microstructural changes in the considered pull-out specimens represent a densification near the reinforcement bar surface and increased porosity within the bulk material. The extent of these changes varies depending on the material composition and thermal exposure, influencing the bond integrity under high-temperature conditions.

The previous study [13] investigated the effect of temperature on the reinforcement bar embedded in refractory concrete. The results indicated that the microstructure changes depend on the reinforcement bar location. The relatively high thermal conductivity of the steel bar regarding the refractory material causes these changes since the reinforcement bars were not protected during the pulling-out specimens’ heating process. Thus, near the contact surface with the reinforcement bar, the material becomes more compact, while the inner regions of the castable exhibit a more porous and chaotic crystal distribution. These microstructural alterations contribute to bonding resistance degradation in several ways. The densification near the reinforcement bar may modify the mechanical interlocking properties. At the same time, the increased porosity inside the castable weakens the overall structural integrity, reducing its resistance to thermal and mechanical stresses. However, these effects may not be apparent when the concrete cover protects bars in real-world structural applications. Still, thermal expansion induces interfacial stresses, proportional to the heat gradient and temperature expansion differences in the steel bar and refractory matrix. These stresses lead to cracking (e.g., Figure 7b) and reduce the bond strength (e.g., Figure 3). Oxidation and corrosion of the reinforcement bar further contribute to the degradation of bonding resistance over prolonged exposure.

At the same time, the combination of the stainless-steel bars and microsilica-modified castable ensures promising bonding resistance. The outstanding mechanical strength of this castable after high-temperature treatment (Figure 2a) prevents premature failure of the bond observed in previous tests in the castable without microsilica [12,13]. Figure 10 shows the stainless-steel bar surface after being pulled from the CC_m_ specimen treated at 1000 °C. This SEM image demonstrates the remaining castable on the pulled-out bar surface after 1000 °C. It indicates sufficient strength of the contact zone between the reinforcement and the castable and the concrete failure controls the bonding resistance. This observation also substantiates the perfect bond assumption of the FE model.

### 4.3. Further Research

Following the composite development concept [37], the FE model created and experimentally verified in this study serves as the reference for assessing the mechanical efficiency of the reinforced refractory composite. In other words, the test outcome of an enhanced material must outperform the numerical prediction result, making the proposed model pivotal for further developing refractory reinforced composites, which have no alternative or adequate reference for estimating their mechanical performance.

The limited dimensions of the bonding zone (Figure 1b) and direct exposition of the reinforcement bars to the high-temperature treatment [12,13,19] may distort the bonding conditions of the reinforcement bars. Therefore, further research should verify the proposed modeling concept for more realistic structural elements. The standardized bending specimens [72] are promising for further analysis.

Because of the refractory material’s complex stress–strain behavior, the full-scale model cannot ensure the reduction in bonding resistance, reducing the concrete strength and deformation modulus. Thus, the model may assume a limited area around the reinforcement bar to represent the bond failure employing the fracture mechanics principles [44] and the characteristic parameters identified in this study. Still, the dimensions of this contact zone affect the finite element mesh and determine the object for further research.

## 5. Conclusions

This numerical study continues the extensive experimental campaign to develop a mechanically efficient structural composite resilient to high temperatures and mechanical loads. The fire and explosion protective structures determine the anticipated application of the developing composite. The previous tests determined the promising composition of the refractory materials. The conducted numerical study develops a simplified model to predict the global bonding behavior of reinforcement bars in the selected refractory material. It offers an empirical equation to mimic the adverse temperature effect by reducing the deformation modulus of the refractory material. The following conclusions result from this numerical study:

The castable’s compressive strength controls the bonding resistance of reinforcement. For the selected materials, the geometry of the test specimens, and simplified modeling assumptions (i.e., the perfect bond model), the maximal bond stress $\tau_{max}=2.5\cdot \sqrt{f_{c}}$, where *f_c_* is the castable’s compressive strength, determine the bonding resistance of the refractory composite. Still, this parameter alone is insufficient for the bond model.The ASTM standard methodology (ASTM E1876-09) is adequate for estimating the selected castable’s deformation modulus, using the ultrasonic pulse velocity (UPV) measurements. These estimations can be used for the chosen castable’s modeling. However, the bonding resistance of steel bars in refractory composite decreases after high-temperature treatment. Therefore, this study proposes the empirical equation for the deformation modulus of concrete to represent the adverse temperature effect on the bonding resistance.Despite the simplicity of the modeling assumptions, i.e., the deformation modulus of the concrete simulates the adverse temperature effects on the bonding resistance, the proposed model adequately predicted the mechanical resistances of the pull-out tests. Independently of the heating temperature, the average pull-out energy prediction error does not reach 4%. The standard deviation of the pull-out deformation energy, assessed during the tests, varies from 11.7% to 22.4% depending on the heating temperature with the 16.2% average value. Thus, the obtained model is considered experimentally verified and reliable for estimating the deformation modulus of the selected refractory castable reinforced with the stainless-steel ribbed bars.This model may serve as a reference for assessing the mechanical efficiency of the reinforced refractory composite: the experimental outcome of a new (enhanced) material must outperform the numerical prediction result to be considered efficient.Further research is necessary to experimentally verify the proposed model’s predictions of the mechanical resistance of structural elements with extended bonding lengths subjected to high-temperature treatments.

## Figures and Tables

**Figure 1 materials-18-01282-f001:**
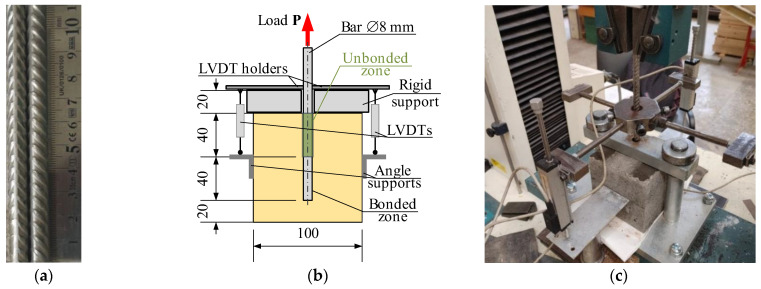
Experimental specimens: (**a**) reinforcement 8 mm bars; (**b**) pull-out sample schematic; (**c**) pull-out test setup.

**Figure 2 materials-18-01282-f002:**
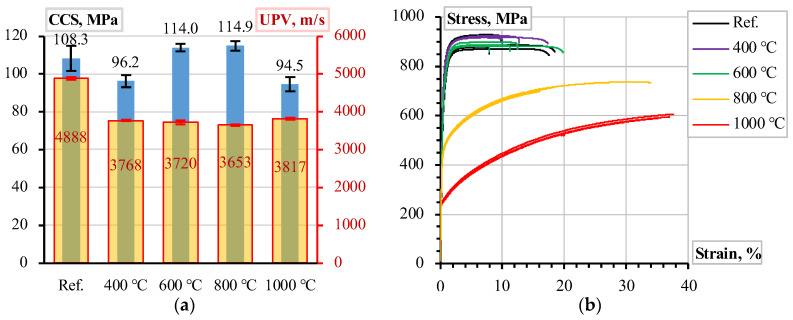
Material characterization results after temperature treatments [12,13]: (**a**) cold compressive strength (CCS) and ultrasonic pulse velocity (UPV) of CC_m_; (**b**) tensile stress–strain diagram of 8 mm ribbed stainless-steel bars.

**Figure 3 materials-18-01282-f003:**
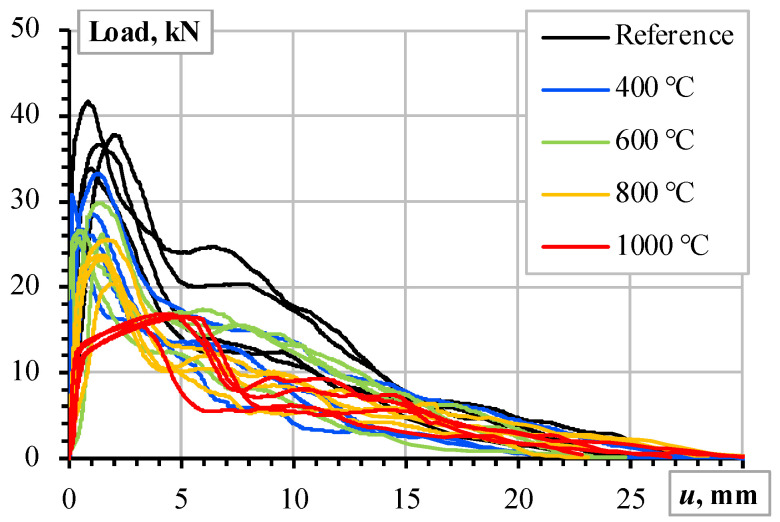
The pull-out test results [13]: the load–displacement relationships of the ribbed stainless-steel bars in CC_m_.

**Figure 4 materials-18-01282-f004:**
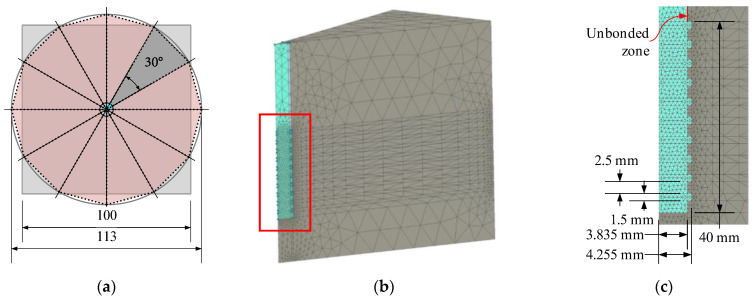
FE modeling schematics: (**a**) approximating pull-out sample; (**b**) FE model of sample fragment; (**c**) detailed view of ribbed bond contact. Note: red rectangle in Figure 4b indicates zoomed part in Figure 4c.

**Figure 5 materials-18-01282-f005:**
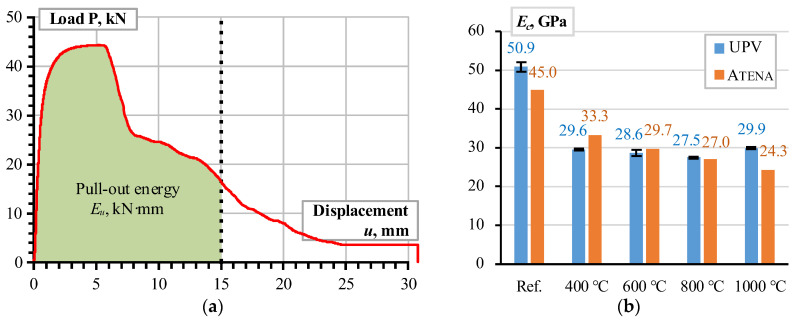
Analysis assumptions and results: (**a**) pull-out energy concept; (**b**) deformation modulus of CC_m_ estimated using UPV measurements and pull-out energy estimation results.

**Figure 6 materials-18-01282-f006:**
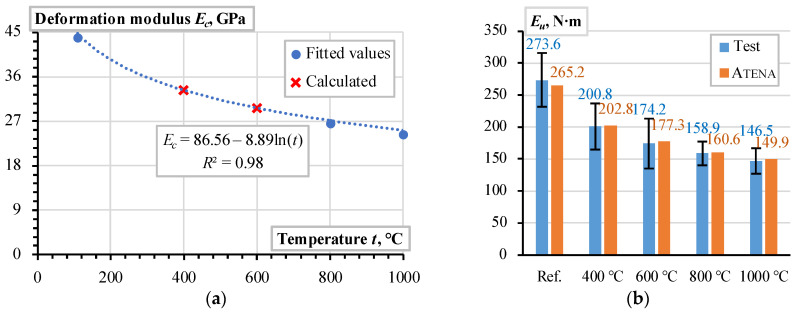
CC_m_ analysis results: (**a**) deformation modulus model; (**b**) pull-out energy assessment.

**Figure 7 materials-18-01282-f007:**
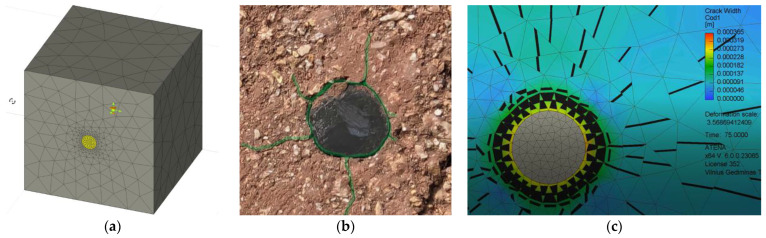
Analyzing uneven expansion of 8 mm steel bar and CC_m_ after treatment at 1000 °C: (**a**) FE model; (**b**) experimental view; (**c**) crack width predictions.

**Figure 8 materials-18-01282-f008:**
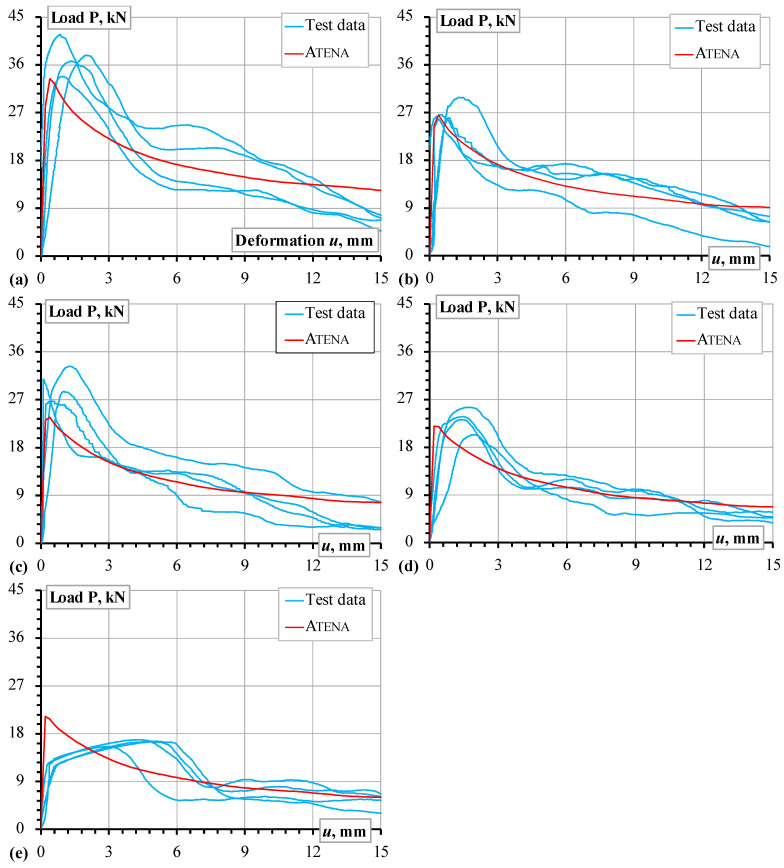
The pull-out modeling results: (**a**–**e**) correspond to the reference samples and specimens treated at 400 °C, 600 °C, 800 °C, and 1000 °C. Note: the test results are the same as in Figure 3.

**Figure 9 materials-18-01282-f009:**
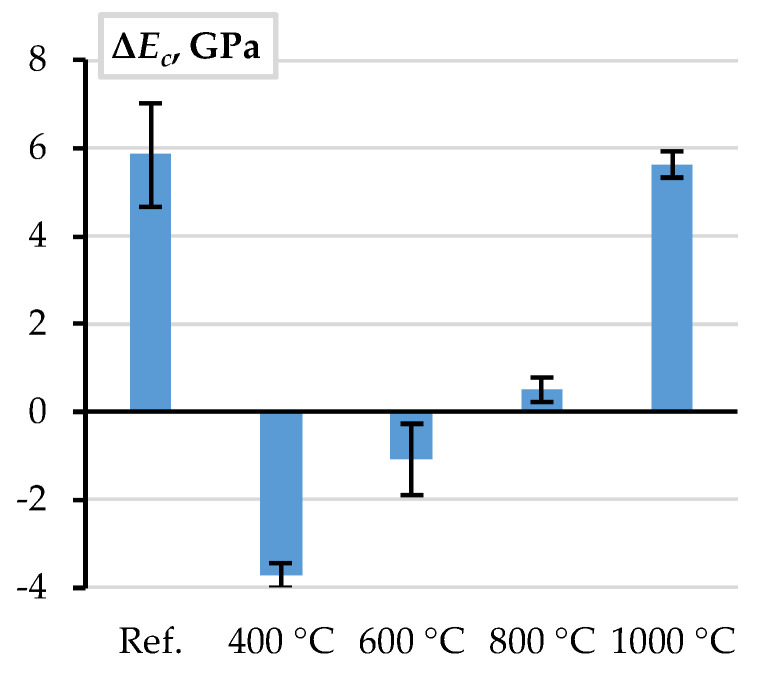
The difference between the deformation modulus of CC_m_, estimated using the UPV measurements, and the pull-out energy estimation results shown in Figure 5b.

**Figure 10 materials-18-01282-f010:**
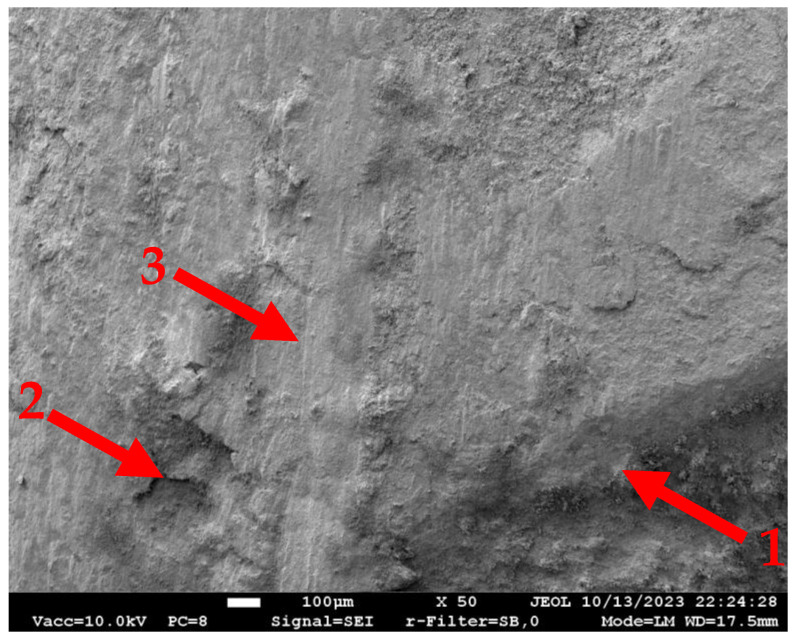
The SEM image (at ×50 magnification) of the bar surface near the rib after the pull-out test of the sample treated at 1000 °C. Note: “1” = reinforcement rib edge; “2” = the castable remained at the bar surface; “3” = the pull-out vertical signs at the sample surface.

**Table 1 materials-18-01282-t001:** Material characteristics assumed for numerical simulations.

Parameter	Value [Unit]	Description
Castable		
*E_c_*	Variable in this study	Modulus of elasticity
*ν_c_* *	0.17 [–]	Poison’s ratio
*f_t_* ^⁑^	3.0 [MPa]	Tensile strength
*G_F_* ^⁑^	75 [N/m]	Fracture energy
*f_c_* ^⁂^	25.5 [MPa]	Compressive strength
*f_c_*_0_ ^⁑^	3.194 [MPa]	Onset of nonlinear behavior in compression
*ε_cp_* ^⁑^	0.00067 [–]	Plastic strain at *f_c_*
*β* *	0 [–]	Parameter controlling the return direction during the concrete crushing
*w_d_* *	0.5 [mm]	Critical displacement in compression
*c_fix_* *	0.7 [–]	Coefficient controlling the switch from rotated to fixed crack model
*c_c_* *	0.05 [m]	Crack spacing
*c_ts_* *	0.4 [–]	Tension stiffening parameter
*ρ_c_*	2308 [kg/m^3^]	Density
*d_a_*	5 [mm]	Maximum aggregate size
*α_c_* ^‡^	0.62 × 10^–5^ [1/°C]	Thermal expansion coefficient (at 1500 °C)
Reinforcement		
*E_s_*	172.2 [GPa]	Modulus of elasticity
*ν_s_* *	0.27 [–]	Poison’s ratio
*f_y_*	623.5 [MPa]	Yield strength
*f_u_*	859.3 [MPa]	Tensile strength
*ε_u_*	0.1 [–]	Rupture strain
*ρ_c_* *	7930 [kg/m^3^]	Density
*α_s_* ^‡^	2.13 × 10^–5^ [1/°C]	Thermal expansion coefficient (at 1000 °C)
Bond model		
Perfect bond	–	No slip allowed through the bonded contact

* Default value; ^⁑^ calculated for assumed compressive strength; ^⁂^ tailored to reflect experimental resistance; ^‡^ values taken from literature [67,68].

## Data Availability

The original contributions presented in this study are included in the article. Further inquiries can be directed to the corresponding author.

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
