# Peer review of "Analyzing the Bonding Resistance of the Ribbed Stainless-Steel Bar in the Refractory Castable After High-Temperature Treatment"

_materials, 2025, doi:10.3390/ma18061282_

Round 1
Reviewer 1 Report
Comments and Suggestions for Authors
The article is on a current and relevant topic, important for the sector and of great interest to the community. The article deals with integrating ribbed stainless Type 304 steel bars with the modified castable to develop a model and this model forms a benchmark for further enhancing the mechanical performance of reinforcement systems (composites) under combined temperature and mechanical loads, diminishing the necessity of expensive experimental trials.
The introduction is quite broad, showing the importance of the research and the gap it seeks to fill, carrying out an extensive bibliographic review.
However, in item 2, Experimental Database, it is important to detail how the 4 heat treatment levels were performed (400 °C, 600 °C, 800 °C, and 1000 °C). How was the heating? How long was the holding time? What was the cooling?...
In item 3, Finite Element Model of the Pull-Out Test, it is important for the authors to emphasize that the data that fed the model came from several authors, from several locations, at different times and even with different boundary conditions. No matter how much one tries to standardize the input data, there is always some error in the work, which must be taken into account when applying the computational model.
And the Discussion of the Results item can be discussed further, in more depth. There is only a list of the main results, without a more critical analysis. It is suggested that all the properties be listed simultaneously in this item.
Author Response
Note. The Authors sincerely appreciate the favorable evaluation of this manuscript. This reply includes only critical remarks, which considerably improved the presentation quality. Yellow highlights all the corrections in the text.
Comment 1. …item 2, Experimental Database, it is important to detail how the four heat treatment levels (400 °C, 600 °C, 800 °C, and 1000 °C) were performed. How was the heating? How long was the holding time? What was the cooling?...
Answer. The Authors appreciate this comment. They intended to avoid repeating the descriptions published in the open-access [12,13,19]. However, recognizing the fragmentary description without these essential parameters, they extended the description of the previous test program. Moreover, this eminent Reviewer’s second comment highlights the apparent gaps in the test program description in the first version of the manuscript.
Correction in the manuscript. The required explanations were added in Lines 140–151 of the updated manuscript. The descriptions of Section 2 were also extended.
Comment 2. In item 3, Finite Element Model of the Pull-Out Test, the authors must emphasize that the data that fed the model came from several authors, from several locations, at different times, and even with different boundary conditions. No matter how much one tries to standardize the input data, there is always some error in the work, which must be considered when applying the computational model.
Answer. This comment is understandable and results from the previously identified presentation gaps. Remarkably, all test results in this manuscript came from the same comprehensive test program conducted by the same Authors. The precisely controlled manufacturing and testing conditions and the quality of the raw materials reduced the procedure flaws and scatter of the results. For instance, the experimental results used for the numerical modeling (Figures 3 and 8, of the updated manuscript) demonstrate the reasonable scatter, which does not contradict the analogous results of the pull-out tests reported in the literature, e.g., [38]. The standard deviation of the experimental pull-out deformation energy assessments (Figure 6b) varies from 11.7% to 22.4% depending on the heating temperature with the 16.2% average value. These results are typical for cementitious composites [71]. Thus, the Authors consider the obtained model reliable for estimating the deformation modulus of the selected refractory castable CCm reinforced with the stainless Type 304 steel ribbed bars. The corresponding comment was added in Section 4, discussing the modeling results. The authors hope this comment and clarifying the test results’ origin (see the reply to Comment 1 above) elucidate the issue.
Correction in the manuscript:
- The last paragraph of the Introduction clarifies the research idea of this study.
- The updated Section 2 adds the relevant comment to clarify the database composition.
- The text on Lines 341–349 of the updated manuscript discusses the quality of the modeling results.
Comment 3. The Discussion of Results item can be discussed in more depth. It only lists the main results without a more critical analysis. It is suggested that all the properties be listed simultaneously in this item.
Answer. This comment was accepted, extending Section 4 and providing particular values and prediction ranges in this section and Conclusions.
Correction in the manuscript. Discussion of the results (Section 4) was substantially extended. The updated section includes three subsections, discussing the modeling adequacy, the bond deterioration/resistance mechanisms, and future research.
Reviewer 2 Report
Comments and Suggestions for Authors
Dear Authors, I think it's good work. However, the following notes and modifications are suggested:
1. The paper assumes a perfect bond model in the numerical simulation. Could the authors discuss the limitations of this assumption and its potential impact on model accuracy?
2. Could the authors provide more insights into the observed trends in the effect of temperature on bond strength and deformation modulus?
3. Were there significant variations in pull-out energy among identical specimens? If so, how were these addressed?
4. What are the primary microstructural changes in the castable after heating, and how do they contribute to bond performance degradation?
5. How do the numerical predictions compare to other existing models in the literature? Have the authors considered benchmarking their model against alternative approaches?
6. How does the proposed model apply to larger-scale structural elements beyond the tested specimens?
7. Were there any observed failure modes in the tested specimens that were not captured by the numerical model?
8. Consider briefly mentioning the key results (e.g., bond performance trends and numerical model validation) to provide a clearer summary.
9. Ensure all figures are clear and well-labeled. For example, Figure 4 could benefit from a more explicit description of key trends.
10. Some terms, such as "bonding integrity" and "bonding resistance," are used interchangeably. Consider defining key terms for consistency.
11. The reference list is comprehensive, but ensure all citations are formatted consistently according to the journal guidelines.
The English could be improved to more clearly express the research.
Author Response
Note. The Authors sincerely appreciate the favorable evaluation of this study. They have reacted to all comments, which considerably improved the presentation quality. Yellow highlights all the corrections in the text.
Comment 1. The paper assumes a perfect bond model in the numerical simulation. Could the authors discuss the limitations of this assumption and its potential impact on model accuracy?
Answer. The Authors acknowledge this note. The perfect bond model’s simplicity determines the essential prominence of this modeling concept, which prohibits slip through the materials’ contact. Alternative approaches form the slip over the inter-material contact through contact elements, empirical models, or other simulation means. In any case, these elaborate models require specific parameters, which experimental determination is not straightforward because of the local slip nonlinearities. Moreover, the so-called bond-slip model complicates the iterative solution process, raising the simulation costs and limiting its practical value.
Remarkably, previous numerical studies, which employed the Atena software, proved the adequacy of the perfect bond model for modeling reinforced concrete cracking and failure. Thus, this study also uses the perfect bond model to simplify the simulation concept typical for engineering applications. The Authors added the corresponding discussion to the updated manuscript.
Correction in the manuscript:
- The text on Lines 208–218 substantiates the prefect bond model choice.
- The text on Lines 404–412 and a new SEM image (Figure 10) provides experimental evidence to substantiate this assumption.
Comment 2. Could the authors provide more insights into the observed trends in the effect of temperature on bond strength and deformation modulus?
Answer. This manuscript does not intend to develop a versatile bond model. This numerical study and previous physical tests [12,13,19] belong to the research project that creates the refractory composite, which combines CCm and ribbed stainless Type 304 steel bars. Typically, the refractory castables have no structural reinforcement and no bond-resistance models. This study is dedicated to filling this modeling gap. Figure 5b of the updated manuscript shows that the UPV counts may reasonably estimate the deformation modulus and approximate the modeling parameters. The temperature expansion deteriorates the bonding performance, and Figure 7b illustrates the deterioration (splitting cracks) caused by the temperature-induced expansion of the steel bar. The Authors also added Figure 9 to show the differences between the estimated experimental and simulated deformation modulus values.
Correction in the manuscript:
- The corresponding comment was added in Lines 327–330.
- New diagram (Figure 9) shows the differences between the estimated experimental and simulated deformation modulus values; and the text in Lines 366–370 discusses the modulus alteration trends.
Comment 3. Were there significant variations in pull-out energy among identical specimens? If so, how were these addressed?
Answer. The standard deviation of the pull-out deformation energy of nominally identical specimens has varied from 11.7% to 22.4% depending on the heating temperature with the 16.2% average value. The black whiskers indicate the standard deviation of the test results in Figures 6b.
Correction in the manuscript: The Authors added the corresponding clarification, discussing these results on Lines 341–349.
Comment 4. What are the primary microstructural changes in the castable after heating, and how do they contribute to bond performance degradation?
Answer. Refractory castables undergo microstructural transformations due to dehydration and phase transformations under high temperatures. As water is expelled from the hydrated phases, the material experiences shrinkage, increased porosity, and cracking. These changes compromise the castable’s overall mechanical integrity and mechanical resistance. The primary microstructural changes in the considered pull-out specimens represent a densification near the reinforcement bar surface and increased porosity within the bulk material. The reinforcement bars’ relatively high thermal conduction regarding the refractory material causes these changes since the reinforcement bars were not protected during the heating process. Thus, near the contact surface with the reinforcement bar, the material becomes more compact, while the inner regions of the castable exhibit a more porous and chaotic crystal distribution. These microstructural alterations contribute to bond performance degradation in several ways. The densification near the reinforcement bar may modify the mechanical interlocking properties. At the same time, the increased porosity inside the castable weakens the overall structural integrity, reducing its resistance to thermal and mechanical stresses. The extent of these changes varies depending on the material composition and thermal exposure, influencing the stability of the bond under high-temperature conditions.
Furthermore, differences in thermal expansion between the steel bar and refractory matrix generate interfacial stresses, leading to microcracking and a reduction in bonding strength. Additionally, oxidation and potential corrosion of the reinforcement bar further contribute to bond degradation over prolonged exposure.
At the same time, the combination of the stainless-steel bars and microsilica-modified castables demonstrates a promising bonding resistance and determines the modeling object in this study. A new SEM image (Figure 10) shows the remaining castable on the pulled-out bar surface after the 1000 °C treatment. It indicates sufficient strength of the contact zone between the reinforcement and the castable and the concrete failure controls the bonding performance. This observation also substantiates the perfect bond assumption of the numerical model.
Correction in the manuscript. Section 4.2 includes the corresponding discussion, the new SEM image, and the bond deterioration mechanisms’ analysis, proving the perfect bond assumption of the model.
Comment 5. How do the numerical predictions compare to other existing models in the literature? Have the authors considered benchmarking their model against alternative approaches?
Answer. The Authors appreciate this comment. Replying to Comment 2 (above), the Authors substantiated the limited knowledge on the bonding performance of the refractory materials operating without reinforcement in typical conditions. Developing a versatile bond model is beyond this study’s scope, which focuses on creating the reference model of the selected refractory components. Thus, the proved suitability of both, UPV counts for estimating the deformation modulus of the castable [65] and the shear strength approach [62,63], in the developed model of the refractory composite determines the essential points to prove the modeling adequacy.
Correction in the manuscript. The answer to Comment 2 describes the corresponding modifications of the manuscript.
Comment 6. How does the proposed model apply to larger-scale structural elements beyond the tested specimens?
Answer. The Authors sincerely appreciate this insightful comment. Because of the complex stress-strain behavior of the material, the full-scale model does not allow for the reduction of castable strength. Thus, the model could assume a limited area around the reinforcement bar to represent the bond failure employing the fracture mechanics principles [44] and the characteristic parameters identified in this study. Still, the dimensions of this contact zone affect the finite element mesh and determine the object for further research.
Correction in the manuscript. The corresponding comment was added describing the prospects for further research (Lines 429–434).
Comment 7. Were any observed failure modes in the tested specimens that the numerical model did not capture?
Answer. The typical consequences of the pull-out tests include the pull-out of the bar (successful outcome since it ensures determining the bonding performance), failure of the reinforcement bar (unsuccessful outcome proclaiming unacceptable long the bonding length), and splitting failure of the concrete cube (unsuccessful outcome proclaiming irrelevantly small dimensions of the cube sample). The two latter consequences were faced in the comprehensive pull-out test program [12,13,19] with other castable and reinforcement materials combinations. On the contrary, the constituent combination (the ribbed stainless-steel bars and microsilica-modified castable) ensured the first (successful) type outcomes for all tests. Therefore, these materials were chosen to develop the refractory composite (in general) and modeling (in this manuscript).
Comment 8. Briefly mention the key results (e.g., bond performance trends, and numerical model validation) to provide a more explicit summary.
Correction in the manuscript. The recommended summary was added in Section 5.
Comment 9. Ensure all figures are clear and well-labeled. For example, Figure 4 could benefit from a more explicit description of key trends.
Answer. Figure 3 (formerly Figure 4) shows the load-deformation diagrams typical for pull-out tests, e.g., [38,58], and barely functional for the trend analysis because of the absence of distinct parameters and substantial scatter. These factors also restrict the quantitative analysis of the model adequacy, using the results of Figure 9. Therefore, this study introduces the pull-out deformation energy measure to estimate the bonding performance in aggregate and “smear” the scatter from the maximum resistance value to the total work released during the bar pull-out. At the same time, the Authors appreciate this note and agree that the original presentation was unclear at least partially. Therefore, the manuscript was carefully proofread and detailed graphical information was commented.
Correction in the manuscript:
- Figure 1 was deleted as uninformative.
- Sections 4.1 and 2 discuss the adequacy of the model assumptions and prediction results.
- A note was added in the caption of Figure 8, relating it to the diagrams in Figure 3.
- A new diagram (Figure 9) reveals the deformation modulus estimation trends.
Comment 10. Some terms, such as "bonding integrity" and "bonding resistance," are used interchangeably. Consider defining key terms for consistency.
Answer. The Authors intended to avoid word duplications. However, they agree that the terminology must be ensured consistent. So, they carefully verified the text to prevent misleading formulations and improve the writing style.
Correction in the manuscript. The text was corrected to improve its clarity and unify terminology. Yellow highlights all the corrections in the manuscript.
Comment 11. The reference list is comprehensive, but ensure all citations are formatted consistently according to the journal guidelines.
Answer. The Authors carefully verified the references to satisfy the template requirement. The bibliographic references now include some excessive information, e.g., the full name of the Journals, issue numbers, and doi-indexes. The Authors typically provide this information, and the editorial managers transform it into the final list without problems.
Correction in the manuscript. The Authors verified all references to satisfy the formatting requirements.
Comment 12. The English could be improved to express the research more clearly.
Correction in the manuscript. The Authors verified the text entirely and corrected it to the best of their abilities. Yellow highlights all the corrections in the text.
Reviewer 3 Report
Comments and Suggestions for Authors
The manuscript is very interesting. I have only a few minor comments.
- Two diagrams and some descriptions in Fig. 1 should be enlarged.
- At the end of the Introduction, the purpose of the research should be clearly defined, like: The aim of the presented research was...
- I was unable to find information regarding the number of repetitions in the pull-out tests. Could you please add that?
Author Response
Note. The Authors sincerely appreciate the favorable evaluation of this study. They have reacted to all comments by the eminent Reviewer. Yellow highlights all the corrections in the text.
Comment 1. Two diagrams and some descriptions in Fig. 1 should be enlarged.
Answer. The Authors appreciate this note. However, they deleted this figure after another Reviewer requested it.
Correction in the manuscript. Figure 1 was deleted.
Comment 2. The purpose of the research should be clearly defined at the end of the introduction, such as: The aim of the presented study was...
Answer. The Authors acknowledge this comment and agree that the research idea was unclear.
Correction in the manuscript. The Authors have rephrased the last paragraph of the Introduction to clarify the research concept.
Comment 3. I could not find information regarding the number of repetitions in the pull-out tests. Could you please add that?
Answer. The number of blue (experimental) diagrams in Figure 8 (of the updated manuscript) describes the number of identical specimens. However, the Authors accepted this criticism, understanding that such an explanation was not straightforward. Therefore, they described these numbers in Section 2. In addition, this section was substantially extended by specifying the testing conditions and other information relevant to understanding the modeling principles and resistance mechanisms behind the observed bonding performance.
Correction in the manuscript. The updated Section 2 determines the number of specimens on Line 174.
Reviewer 4 Report
Comments and Suggestions for Authors
The paper studied the bond performance of the ribbed stainless-steel bar in refractory castable after high-temperature treatment. A numerical model was created to predict the bonding performance of the selected material combination. The following problems must be modified.
- The Introduction part has written a lot of tedious introduction to the literature, not refined enough. They should focus on the purpose and significance of this study.
- For Figure 1, as this professional people understand, there is no need to put into the paper.
- How much higher the prediction accuracy of this work is than traditional method?
- This study is a follow-up to the previous articles, and the published data graph should not be reproduced, although it is marked with references, for example, Figures 3 and 4. Just explain in the text, the authors can review your published papers.
- The resolution of Figure 5 is too low to see.
Author Response
Note. The Authors have reacted to all comments from the eminent Reviewer. Yellow highlights all the corrections in the text.
Comment 1. The Introduction contains many tedious introductions to the literature that are not refined enough. It should focus on the purpose and significance of this study.
Correction in the manuscript. The Introduction was refined as requested.
Comment 2. As professionals understand, the paper does not need to include Figure 1.
Correction in the manuscript. This figure was deleted as suggested.
Comment 3. How much higher is the prediction accuracy of this work than that of the traditional method?
Answer. This study develops the refractory composite, which combines castable material and ribbed stainless-steel bars. Typically, these castables have no structural reinforcement and no bond-resistance models are necessary. This study is dedicated to filling this modeling gap and creates the bond model for the selected combination of refractory materials. Figure 5b of the updated manuscript shows that the UPV counts may reasonably estimate the deformation modulus and approximate the modeling parameters. The temperature expansion deteriorates the bonding performance, and Figure 7b illustrates the deterioration (splitting cracks) caused by the temperature-induced expansion of the steel bar. The Authors also added Figure 9 to show the differences between the estimated experimental and simulated deformation modulus values.
In other words, developing a versatile bond model beyond this study’s scope focuses on creating the reference model of the selected refractory components. Thus, the proved suitability of both, UPV counts for estimating the deformation modulus of the castable [65] and the shear strength approach [62,63], in the developed model of the refractory composite determines the essential points to prove the modeling adequacy.
Correction in the manuscript:
- The corresponding comment was added in Lines 327–330.
- New diagram (Figure 9) shows the differences between the estimated experimental and simulated deformation modulus values; and the text in Lines 366–370 discusses the modulus alteration trends.
Comment 4. This study follows up on previous articles, and the published data graph should not be reproduced, although it is marked with references, for example, Figures 3 and 4. Just explain in the text, the authors can review your published papers.
Answer. The Authors appreciate this comment. They intended to avoid repeating the descriptions published in the open-access [12,13,19]. However, recognizing the fragmentary description without these essential parameters and following the requirements of other Reviewers, they extended the description of the previous test program.
This manuscript reports the testing characteristics and experimental results essential for understanding the modeling object. Figures 2 and 3 (former Figures 3 and 4) are mandatory for understanding the simulation object and modeling parameters. For instance, Figure 2a is a brand-new diagram representing the UPV values necessary to calculate the deformation modulus using Equation 1. In addition, this figure is essential to prove the necessity of alternative relationships, which are typical for structural concretes, and to relate the modulus of elasticity and compressive strength values.
Figures 2b and 3 are necessary to understand the temperature effect on the selected refractory composite’s mechanical performance, particularly bonding resistance. Therefore, the Authors have decided not to eliminate these figures from the manuscript.
Correction in the manuscript. Section 2 was slightly extended to clarify the specimen preparation procedures (satisfying other Reviewers’ comments).
Comment 5. The resolution of Figure 5 is too low to see.
Answer. The Authors apologize for the mislaid visualization.
Correction in the manuscript:
- The Authors flipped the view on Figure 4b (formerly Figure 5b) in horizontal direction to ensure the fragment orientation conformed to the images in Figures 4a and 4c, and added the red rectangle, which clarified the location of the zoomed zone.
- The figure caption now includes the corresponding note to clarify the issue.
Round 2
Reviewer 2 Report
Comments and Suggestions for Authors
Dear authors
I would like to thank you for making the necessary modifications.